# In Vivo Characterization of Intracortical Probes with Focused Ion Beam-Etched Nanopatterned Topographies

**DOI:** 10.3390/mi15020286

**Published:** 2024-02-17

**Authors:** Jonathan L. Duncan, Jaime J. Wang, Gabriele Glusauskas, Gwendolyn R. Weagraff, Yue Gao, George F. Hoeferlin, Allen H. Hunter, Allison Hess-Dunning, Evon S. Ereifej, Jeffrey R. Capadona

**Affiliations:** 1Department of Biomedical Engineering, Case Western Reserve University, 10900 Euclid Ave, Cleveland, OH 44106, USA; 2Advanced Platform Technology Center, Louis Stokes Cleveland Veterans Affairs Medical Center, 10701 East Blvd, Cleveland, OH 44106, USA; 3Michigan Center for Materials Characterization, University of Michigan, 500 S. State St, Ann Arbor, MI 48109, USA; 4Department of Biomedical Engineering, University of Michigan, 500 S. State St, Ann Arbor, MI 48109, USA; 5Veterans Affairs Hospital, 2215 Fuller Rd, Ann Arbor, MI 48105, USA

**Keywords:** nanopatterned, microelectrode, neuroinflammation, gene expression, FIB

## Abstract

(1) Background: Intracortical microelectrodes (IMEs) are an important part of interfacing with the central nervous system (CNS) and recording neural signals. However, recording electrodes have shown a characteristic steady decline in recording performance owing to chronic neuroinflammation. The topography of implanted devices has been explored to mimic the nanoscale three-dimensional architecture of the extracellular matrix. Our previous work used histology to study the implant sites of non-recording probes and showed that a nanoscale topography at the probe surface mitigated the neuroinflammatory response compared to probes with smooth surfaces. Here, we hypothesized that the improvement in the neuroinflammatory response for probes with nanoscale surface topography would extend to improved recording performance. (2) Methods: A novel design modification was implemented on planar silicon-based neural probes by etching nanopatterned grooves (with a 500 nm pitch) into the probe shank. To assess the hypothesis, two groups of rats were implanted with either nanopatterned (n = 6) or smooth control (n = 6) probes, and their recording performance was evaluated over 4 weeks. Postmortem gene expression analysis was performed to compare the neuroinflammatory response from the two groups. (3) Results: Nanopatterned probes demonstrated an increased impedance and noise floor compared to controls. However, the recording performances of the nanopatterned and smooth probes were similar, with active electrode yields for control probes and nanopatterned probes being approximately 50% and 45%, respectively, by 4 weeks post-implantation. Gene expression analysis showed one gene, Sirt1, differentially expressed out of 152 in the panel. (4) Conclusions: this study provides a foundation for investigating novel nanoscale topographies on neural probes.

## 1. Introduction

Intracortical microelectrodes (IMEs) are critical tools for recording electrical signals from multiple individual neurons simultaneously. As a component of brain–machine interfaces (BMIs), IMEs record neural signals that can be translated into control signals for assistive technologies for individuals with disabling neurological conditions such as spinal cord injury or amyotrophic lateral sclerosis (ALS), as well as limb loss [1,2,3,4]. By capturing coordinated neural activity across brain regions, IMEs hold a key role in generating insights into the functional connectivity of neural circuits [5,6]. The signals encoded by single unit spikes from individual neurons offer high spatiotemporal precision, which requires placing the sensing microelectrodes within close proximity to healthy, active neurons [7,8]. Silicon-based IME probes can be fabricated using processes developed by the microchip industry and serve as a standard for positioning multiple recording electrode sites within the cortex [9,10].

One of the primary challenges to realizing the potential of intracortical neural probes in basic science and clinical applications is the gradual signal degradation and device failure characteristic of chronic IME studies [11,12,13,14,15]. A primary factor contributing to the decline in recorded signal quality is the neuroinflammatory response resulting from device insertion and continued presence within cortical tissue. Device insertion into the cortex causes vasculature and cellular injury, initiating an immune response involving microglia and macrophage activation and migration to the implant site, pro-inflammatory cytokine release, reactive oxygen species (ROS) production, astrocytic encapsulation and neuronal degradation [16,17,18,19,20]. Recording performance failure, therefore, results from probe encapsulation via glial scar formation, increased distance between the electrode sites and active neurons, as well as oxidation and degradation of the implanted device materials [15,19,20,21,22,23].

Approaches to mitigating the neuroinflammatory response have included the use of soft IME device structural materials to minimize mechanical mismatch [24,25,26], integrating anti-oxidative coatings [27,28,29] or other methods of therapeutic delivery [30,31,32], and using cellular-scale geometric features [33,34,35]. Though the previous approaches for neuroinflammatory response mitigation have demonstrated promising results, they often involve substantial changes in materials and manufacturing schemes compared to existing commercially available intracortical devices [36,37]. Implant modifications that can be applied to current devices may be more readily translated to clinical applications and neuroscience studies.

The extracellular matrix (ECM) of cortical tissue is made of a complex meshwork of 3D protein structures [38], with features in stark contrast to smooth silicon-based neural probes [39]. Several studies have shown that proteins and cells respond to topographical cues [40,41,42,43,44], suggesting implants that mimic the nanoscale ECM topography may mitigate the neuroinflammatory response to the implant. Ereifej et al. compared the neuroinflammatory responses to silicon dummy probes with nanopatterned grooves etched along the shank and controlled smooth silicon dummy probes [41]. Immunohistochemistry (IHC) results indicated a higher concentration of viable neurons within 150 µm of the nanopatterned probes than with the smooth control probes. Additionally, the expression of *Cd14*, a gene associated with innate immune response involved in neuroinflammation [45,46,47,48,49], was lower for the nanopatterned probes than for smooth controls. Here, we sought to build upon our previous work and implement the nanopatterned topography on functional recording neural probes to investigate the effects of nanopatterning on neural recording performance and neuroinflammatory gene expression.

## 2. Materials and Methods

### 2.1. Neural Probe Manufacturing

Customized single-shank, silicon-based 16-channel intracortical microelectrode probes (NeuroNexus, Ann Arbor, MI, USA, A1×16-3mm-100-177-FIB) were purchased from NeuroNexus. An additional step was added to the standard NeuroNexus process to add a layer of gold to the top surface of the probes at the wafer level. This was achieved through sputter deposition of a titanium adhesion layer (20 nm) followed by a sputter-deposited gold layer (300 nm) (Figure 1(A1)). The gold layer began 6 µm from the edge of the probe and had 3–10 µm distances from each recording site. The distance between the gold and the recording sites became progressively larger distances since the device tapers, starting from the tip: Site 1: 3 µm, Site 2: 5 µm Site 3: 8 µm, Sites 4–10: 10 µm. The gold/titanium layers were patterned using lift-off to expose and isolate the recording sites and connector contacts (Figure 1(A2)). Following this, the probes were connectorized and packaged following standard NeuroNexus procedures.

The probes were taken to the Michigan Center for Materials Characterization at the University of Michigan (Ann Arbor, MI, USA) to be etched with specific nanopatterns. Electrodes were prepared for nanogroove patterning by attaching to 1 inch-diameter aluminum stubs using conductive copper tape. A small strip of copper tape was bonded to the proximal end of the probe shank with a small amount of colloidal silver paint (Ted Pella Inc., Redding, CA, USA no. 16032) to provide a conductive path to the silicon probe tip from the aluminum stub (Figure 1B). The copper tape was carefully removed with tweezers after focused ion beam (FIB) etching was complete, but a tiny amount of the silver paint remained at the very proximal end of the electrode shank.

Parallel nanogrooves were etched onto the probe shanks on both the top (gold) and bottom (silicon) sides by FIB using 30 kV Ga ions with an FEI Helios 650 Dual Beam microscope (Hillsboro, OR, USA) at the Michigan Center for Materials Characterization at the University of Michigan (Ann Arbor, MI, USA) (Figure 1(A3,4)). A beam current of 2.5 nA was used to etch lines with a pitch of 500 nm into the surface, as seen in Figure 2. The etch pattern was controlled using Nanobuilder 2.0 software (ThermoFisher, formerly FEI Co., Hillsboro, OR, USA). The pattern was designed such that all FIB etching took place outside of a distance of 3–10 µm from each recording site.

As the length of the probes exceeded the maximum field of view of the FIB, the length of the shank was divided into five blocks that were etched sequentially, with adjustments to the sample and stage position between blocks. After etching one side of the probe, the probe was removed from the microscope, flipped over, and the opposite side was then etched.

### 2.2. Animals Surgeries

All animal care, handling and procedures were performed in compliance with a protocol approved by the Institutional Animal Care and Use Committee (IACUC) at Case Western Reserve University. Twelve 10-week-old Sprague–Dawley rats (Charles River Labs, Wilmington, MA, USA) were implanted either with one “smooth” control probe (n = 6) or one nanopatterned probe (n = 6). Implants were sterilized using ethylene oxide gas at 54.4 °F, with 1 h sterile time and 12 h aerate.

Once the animals were anesthetized with isoflurane (3.5–4.0%), the fur on the scalp around the surgical site was removed, the toenails were clipped and eye ointment (Systane Ointment, Alcon Canada Inc., Mississauga, ON, Canada) was applied. The surgical site was prepared by scrubbing the surgical area with betadine and 70% isopropanol. The animal was then placed on a warming pad (PhysioSuite^®^, Kent Scientific Corporation, Torrington, CT, USA) and secured within a calibrated digital stereotaxic frame (David Kopf Instruments, Tujunga, LA, USA), ensuring precise and reproducible placement of the rodent’s head. Lidocaine (10 mg/kg) was administered subcutaneously to the incision site and buprenorphine (0.05 mg/kg) was administered subcutaneously on the back of the neck.

A one inch midline incision was made on the top of the head; the skin was retracted, and the periosteum was removed to expose the skull. Hydrogen peroxide was applied to dry the skull surface and expose the bregma and lambda landmark points for craniotomies. The skull surface was primed with a drop of Vetbond tissue adhesive (Catalog #70200742529, 3M, Saint Paul, MN, USA) to enhance dental cement adhesion to the skull.

To target the primary motor cortex (M1), one craniotomy was established 2 mm to the right laterally and 2 mm anterior to the bregma using a stereotaxic frame-mounted dental drill with a 1.75 mm drill bit (Stoelting Co., Wood Dale, IL, USA). Two auxiliary craniotomies were created using a 1.35 mm drill bit (Stoelting Co., USA) for the insertion of the ground and reference screws (stainless steel bone screws, Stoelting Co., Wood Dale, IL, USA), hand-drilled into the skull turning only 2.5 times to prevent touching brain tissue, both positioned 1.5 mm to the left lateral of the midline and 1.5 mm (ground) and 5.5 mm (reference) posterior to the bregma. A final craniotomy was created using a 1.35 mm drill bit for an anchoring screw (1.5 mm lateral and 3 mm posterior to bregma) to help secure the dental cement to the skull following established protocols [50].

After dura removal, a control “smooth” or nanopatterned 16-channel single-shank intracortical microelectrode probe was positioned above the craniotomy made anterior to the bregma, aligning with the exposed cortical tissue. The reference and ground wires were secured in place by wrapping around the screws. The probe was inserted into the motor cortex to a depth of 2 mm using a NeuralGlider inserter (Actuated Medical, Bellefonte, PA, USA). Kwik-Sil silicone adhesive (World Precision Instruments, Sarasota, FL, USA) was applied to seal the craniotomy, after which Teets Cold Cure dental cement (A-M Systems, Sequim, WA, USA) was then applied to form a cement head cap to secure the implant in place. Several 5-0 monofilament polypropylene sutures were placed on both ends of the incision, ensuring proper healing while still providing access to the electrode connector head stage. Sutures were removed 14 days post-surgery.

Antibiotic therapy comprised a single dose of cefazolin (5 mg/kg) administered immediately post-surgery, followed by continuous administration of trimethoprim sulfamethoxazole (53 mg/kg/24 h) in the animal’s drinking water for 7 days. Pain management included subcutaneous administration of buprenorphine (0.05 mg/kg) administered twice daily for up to 72 h.

### 2.3. Neural Recordings

All recordings were performed twice weekly for 4 weeks. The day of implantation was marked as day 0 to establish baseline recordings. Animals were briefly anesthetized using isoflurane (3.5% with 1.5 L/min O_2_) to allow for making secure connections to the head stage. Connector contacts were cleaned with 70% isopropanol swabs. Animals were placed in a Faraday cage to reduce electromagnetic noise, then connected to a 16-channel ZIF-Clip head stage (Tucker-Davis Technologies Inc., Alachua, FL, USA) attached to a commutator (Catalog #ACO32, Tucker-Davis Technologies Inc.), which allowed the animal to roam around the box without twisting the cable. The commutator was connected to a 16-channel Tucker Davis Technologies Lab Rat digital signal processor (Tucker-Davis Technologies Inc.), and the processed signals were collected by a computer (Dell Technology Company, Round Rock, TX, USA) via a USB 3.0 cable. After the animals woke up, neural signals were recorded for 10 min and visualized using SynapseLite software (Tucker-Davis Technologies Inc.). Data was sampled at 24,414 Hz and bandpass filtered from 300–3000 Hz. The SynapseLite sorting feature was utilized to confirm the presence of single units at the 16 contact sites. One animal in each group failed to record neural signals at any time during the study and was thus removed from the recording analysis (n = 5 per group), though remained in the study for inflammatory markers (n = 6 per group). A general setup of recording and data acquisition can be seen in Figure 3.

### 2.4. Recording Analysis

A Plexon offline sorter (Plexon Neuroscience Technologies, Dallas, TX, USA) was used to determine and confirm the presence of single-unit action potentials at each of the 16 contact sites. Common average referencing across all electrodes was implemented to remove motion artifacts caused by animal actions, such as head movement and grooming. Additionally, spikes with amplitudes exceeding +/− 300 µV and cross-channel artifacts that appeared on at least 14 channels were removed. Waveform settings used 1720 µs for waveform length, a pre-threshold period of 410 µs and a dead time of 1352 µs. Individual spikes were detected using a −4σ standard deviation cutoff from the mean. K-means automated scanning was used to detect and cluster single unit spike events across 16 channels for each of the experimental groups. Manual sorting of the single unit spike events was then performed for all channels and both experimental and control groups.

These metrics were summarized by taking the detected units from each group in Plexon and evaluating their electrophysiological performance using a custom MATLAB R2023a code (Mathworks, Natick, MA, USA) to calculate standard metrics of peak-to-peak voltage, root mean square noise, median spike rate, signal-to-noise ratio and proportion of active electrode yield. The noise level was calculated as the root mean square (RMS) of the recorded and filtered signal after artifact and spike removal. The peak-to-peak voltage (Vpp) was the peak-to-trough potential difference for each spike waveform. The signal-to-noise ratio (SNR) was calculated by dividing the Vpp of each unit by the RMS noise. The median spike rate was calculated as the reciprocal of the interspike interval. The average number of distinct units detected per channel was determined by dividing the total number of identified units by the total number of channels per group. A two-pronged approach was used to determine the significance of the recording metrics: first, a Kruskal–Wallis and, second, a Benjamini–Krieger–Yekutieli test for multiple comparisons with non-normal distributions to reduce type I errors. Different levels of significance were determined in GraphPad using the following notation with *p*-values: * = 0.01–0.05, ** = 0.001–0.01, *** = 0.0001 to 0.001 and **** =< 0.0001. All statistical data was visualized on a GraphPad Prism 10 (GraphPad Software, Boston, MA, USA).

To determine the proportion of active channels, the total number of active channels (any contact site that recorded at least one unit during the week) was summed and then divided by the total number of channels (16 channels per probe for 5 animals for each week), yielding a sample size of n = 80 recording channels on nanopatterned probes and n = 80 channels on control probes, both representing a per week count. With a total of N = 320 sites across 4 weeks for each experimental group, a statistical significance was determined by a one-tailed test of proportions Z-test between patterned and control groups using Microsoft Excel.

### 2.5. Cardiac Perfusions

Animals were anesthetized using isoflurane and then injected with a cocktail of ketamine (140 mg/kg) and xylazine (20 mg/kg). Once sufficient depth of anesthesia was confirmed by toe pinch, the animal was placed and secured in a perfusion tray. A small cut was made around the xiphoid process to remove the skin, and then another cut was made to expose the pleural cavity. Cuts were made across the abdominal cavity, vertically through the ribs, and finally across the diaphragm. The sternum flap was clamped with a hemostat and flipped over the animal’s head to expose the heart. The left ventricle was snipped, and a metal gavage needle was placed through the incision and up into the aorta and clamped into place with a hemostat. The right atrium was then clipped to allow for blood and perfusate to drain from the animal. Next, 100–250 mL each of 1× phosphate buffer saline (PBS), followed by 30% sucrose (Millipore Sigma, Burlington, MA, USA) in 1× PBS was perfused through the heart at 20–35 RPM using a Golander peristaltic pump (BT600S, Golander LLC, Norcross, GA, USA). Upon completion of perfusion, the head was decapitated, and the brain was removed from the skull and placed in a container filled with Optimal Cutting Temperature (OCT) compound (Sakura Finetek USA Inc., Torrance, CA, USA, #25608-930), then flash frozen in dry ice and stored at −80 °C.

### 2.6. RNA Isolation

To collect tissue for RNA isolation, the brains were cryosectioned with a Leica CM1860 Cryostat (Leica Biosystems, Deer Park, IL, USA). OCT was trimmed until the implant sites were visible. Ten 150 µm horizontal slices of the brain were then taken along the length of the probe, and a 1 mm diameter biopsy punch (CAT#12-460-402, Fisher Scientific, Hampton, NH, USA, Integra Militrex Biopsy Punch w/Plunger) was centered over the implant site and used to extract the tissue surrounding the implant site. The extracted tissue was pooled with all ten slices from the same animal and placed in a 1.4 mm Omni nuclease-free bead tube (CAT# 19-627, Omni International, Kennesaw, GA, USA), then stored at −80 °C until RNA isolation.

RNA isolation was performed via the QIAsymphony SP and QIAsymphony RNA kit (CAT#931636, Qiagen, Hilden, Germany) according to manufacturer protocols (RNA_CT_400_V7, Qiagen, Hilden, Germany). First, a buffer RLT solution was made with 400 µL Buffer RLT Plus (QIAsymphony RNA kit, Qiagen, Hilden, Germany) per sample, 10 µL 2-Mercaptoethanol (CAT#O3446I-100, Fisher Scientific, Hampton, NH, USA) per 1 mL Buffer RLT Plus, and 0.5% *v*/*v* of Reagent DX (CAT#19088, Qiagen, Hilden, Germany). Next, 400 µL of buffer RLT solution was added to each sample bead tube, and the tissue was disrupted and homogenized via Omni Bead Ruptor 12 (SKU 19-050A, Omni International, Kennesaw, GA, USA). Lysate was centrifuged for 3 min at 14,000 rpm, and the supernatant was transferred to 1.5 mL Eppendorf tubes. 100 µL of chloroform was added to each sample supernatant, and the sample was vortexed thoroughly for 3 s and centrifuged at 4 °C for 3 min at 14,000 rpm. The top layer, the final lysate, was then transferred to 2 mL conical tubes (CAT#997102, Qiagen, Hilden, Germany). Lysate tubes were loaded into the sample drawer of the QIAsymphony SP along with the required reagents, consumables, and eluate cooling adapter and tubes. The instrument then performed the automated extraction of RNA, which was eluted at 50 µL into 2 mL conical tubes (CAT#997102, Qiagen, Hilden, Germany). Eluted RNA was stored at −80 °C until measurements were taken with the NanoString nCounter^®^.

### 2.7. Gene Expression Assay

Genomic analysis was conducted via a NanoString Technologies’ nCounter^®^ Analysis System (Seattle, WA, USA). RNA samples were hybridized with a reporter codeset and capture codeset. The reporter codeset contains fluorescent probes for a custom panel of 152 genes of interest (Table 1) [27,50,51,52,53,54], including six housekeeping genes (HK). The panel was selected based on significant genes from our previous studies and available oxidative stress markers [41,51,55,56]. Negative and positive controls were spiked in. Samples were incubated at 65 °C for 18 h and then loaded into the nCounter^®^ Prep Station for sample processing into cartridges. Cartridges were then loaded into the nCounter^®^ Digital Analyzer, while probe counts were read at 280 Field-of-View per sample.

### 2.8. Data Visualization and Statistical Analysis

Genes with counts below 20 in over 85% of samples were excluded from analysis. Raw gene counts were normalized by positive controls and HK genes. Differential expression was calculated by taking the average of the normalized counts per gene of the patterned electrodes and dividing by the respective average in controlled electrodes. The expression ratio was plotted on a log2 scale as log2FoldChange (log2FC). Statistical significance was calculated with a two-tailed *t*-test for unequal variance with a Benjamini–Hochberg False Discovery Rate correction to determine the adjusted *p*-value (padj), with significance set at padj < 0.1. All six animals from each group were included in the analysis of gene expression.

Data was visualized through volcano plots made in GraphPad, with log2FC on the x-axis and −log10(padj) on the y-axis. Genes on the left side of the plot (log2FC < 0) showed greater expression in the control group. Genes on the right side (log2FC > 0) showed greater expression in the nanopatterned group. Higher up in the plot indicates greater significance.

## 3. Results

### 3.1. Recording Metrics

Prior to implantation, electrode site impedance values were measured before and after FIB etching (Figure 4). Violin plots of all respective values before and after FIB demonstrate an increase in impedance for some of the contacts following FIB treatment, which is likely attributed to FIB-induced damage to microelectrode contacts. The large range of post-FIB impedance values correlates with the increasing misalignment in patterning along the length of the probe (Figure 2). Microelectrode contacts toward the tip are well-aligned, with all FIB exposure occurring outside of the contact. Toward the base, the slight rotational misalignment is magnified, resulting in FIB patterning on the microelectrode contacts. Since the thickness of the contacts is like the depth of nanogrooves, it may be the case that some of the microelectrodes were damaged, resulting in higher impedance values.

The in vivo recording metrics are summarized in bar plots and violin plots in Figure 4. All results comprise the weeks within the acute phase of implantation (1–4 weeks) [13]. One animal from each group showed little to no signal during unit sorting and was removed from the recording analyses. Figure 5A shows that the proportion of active recording electrodes in nanopatterned and control probes was similar throughout the study. In week 1, both groups were recording an active yield performance of 65%. In subsequent weeks, the signal quality, and the ability to detect signals in the nanopatterned probe group, deteriorated at statistically similar rates to the control probe group. By the end of the study, at 4 weeks, approximately 50% of the control probe recording electrodes and 45% of the nanopatterned probe recording electrodes were active (Figure 5A). Spikes detected by the control probes had a significantly higher signal-to-noise ratio at weeks 2 and 4 compared to the nanopatterned probes (Figure 5B). As seen in Figure 5C, peak-to-peak voltage (Vpp) was generally between approximately 25 and 55 µV for both groups, except for the nanopatterned probes during week 4. No significant difference in Vpp between groups was detected for any of the weeks investigated in this study. Figure 5D shows that, in week 4, the noise floor was significantly higher in recordings from nanopatterned probes compared to control probes. The average number of distinct units detected per recording electrode was consistently one unit per electrode across groups and time points (Figure 5E). No significant difference in the average number of units per recording electrode was found among the treatment groups. The nanopatterned electrodes for weeks 2 and 4.

### 3.2. Gene Expression Analysis

To understand the neuroinflammatory response around the implant site, we examined the differential gene expression between the control group and the nanopatterned group (n = 6 each). Our results showed that one gene, Sirt1, demonstrated significant differential expression and was downregulated in the tissue surrounding nanopatterned probes compared to smooth control probes. Sirt1 encodes the protein Sirtuin 1 (SIRT1), which suppresses gene expression through histone deacetylation (Figure 6). Sirt1 is involved in the negative regulation of apoptosis and has been seen to be anti-inflammatory by repressing proinflammatory cytokine expression in astrocytes and microglia [57,58,59].

## 4. Discussion

This in vivo study compared the recording performance and differential gene expression of nanopatterned probes compared to control probes. Initial histological results from Ereifej et al. were encouraging, showing that nanopatterned electrodes resulted in lower expressions of nitric oxide synthase and the tumor necrosis factor at 2 and 4 weeks, respectively [41]. However, when applied to probes for functional recording, we found that the nanoscale topography did not lead to improved recording performance (Figure 5A). Recordings from nanopatterned and smooth probe groups showed a gradual signal deterioration from week-to-week, which is consistent with previous studies [14,60]. We found little significant difference in the SNR between control and patterned electrodes, except for weeks 2 and 4, where the control was higher than the patterned (Figure 5B). A similar trend could be seen with respect to the RMS noise seen in Figure 5D, where there was no significance for week 2, while week 4 showed significance for the patterned over control. Together, low SNR and increased noise could be a result of the FIB etching for the patterned electrodes. Figure 4 displays impedance before and after FIB etching, noting that nanopatterned probes have impedances ranging from 1 MΩ to 4 MΩ. An acceptable range for recording range, as mentioned by Cogan, is between 5 kΩ and 1 MΩ [61]. Pancrazio et al. found a more conservative range of 100 kΩ to 900 kΩ acceptable [62]. While surface modification of microelectrodes has been shown to improve neuron survival and promote greater cell adhesion, translation to recording studies has not yielded significant improvements to the same degree as seen in passive electrode experiments.

Regarding our gene expression analysis, we saw significant downregulation of one gene, Sirt1, from tissue adjacent to nanopatterned probes compared to control probes. Although differential expression of a single gene does not permit definitive conclusions about the mechanisms underlying the neuroinflammatory response to smooth vs. nanopatterned probes, this observation warrants further investigation. Because Sirt1 suppresses the expression of pro-inflammatory cytokines, higher expression of Sirt1 near smooth control probes may indicate greater attenuation of inflammation. The corollary suggests that the downregulation of Sirt1 adjacent to the nanopatterned electrodes would be indicative of enhanced neuroinflammation. It would be beneficial to analyze gene expression at prior time points, such as 1 and 2 weeks post-implantation, to assess the change in response over time and determine if more significant differences in neuroinflammation occur at earlier timepoints.

The previous histology study implemented parallel grooves on silicon dummy probes without recording electrode sites, insulating films, or the added gold layer, which may contribute to the discrepancies between the non-recording and recording studies. Though nanostructured gold has been shown to decrease astrocyte coverage while maintaining neuron coverage compared to planar gold in an in vitro study [40,63], implementing nanostructured gold with parallel nanogrooves may not have the same effect. Future studies may investigate differential gene expression at the implant sites to determine the effects of the materials that carry the nanotopography.

## 5. Conclusions

This study established that nanopatterned coatings can be produced by sputter deposition and focused ion beams on silicon-based probes made using existing commercial processes with established designs. The addition of the sputtered gold coating allowed for producing the nanogrooves. However, increased impedance values following FIB etching prior to implantation in rodents do not ensure that FIB etching comes without any damage to the insulating layers or altering of the electrical properties of the established NeuroNexus probes. Importantly, the modified probes demonstrated an ability to record single units to an extent, and with a quality similar to recordings from the unmodified probes. Given that the FIB patterning is software-controlled, the established process for modifying NeuroNexus probes can be used to apply arbitrary nanopatterned features that may influence the morphology and functions of cells surrounding the implant [40,44,64].

## Figures and Tables

**Figure 1 micromachines-15-00286-f001:**
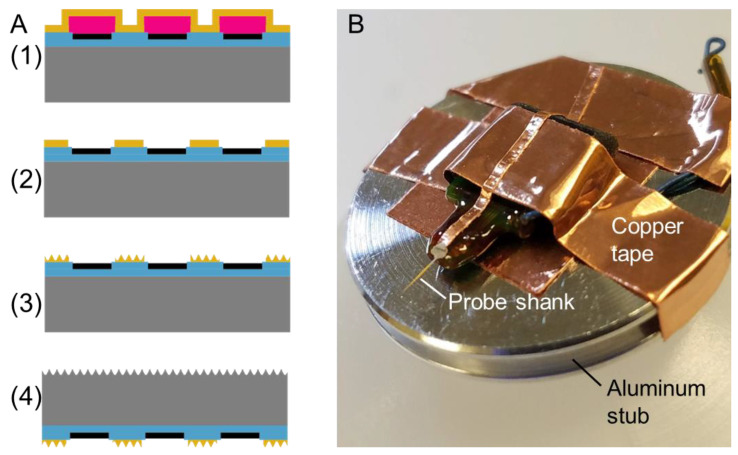
(**A**) Process flow for FIB-patterned probes: (1) titanium and gold were sputter-deposited on the probes prepared with photoresist; (2) Liftoff was used to pattern the titanium and gold layers; (3) nanopatterned topographies were formed into the gold layer using FIB; (4) the probe was flipped over, and the backside was subjected to FIB patterning. (**B**) photogram of probe taped to aluminum stub for FIB work-flow.

**Figure 2 micromachines-15-00286-f002:**
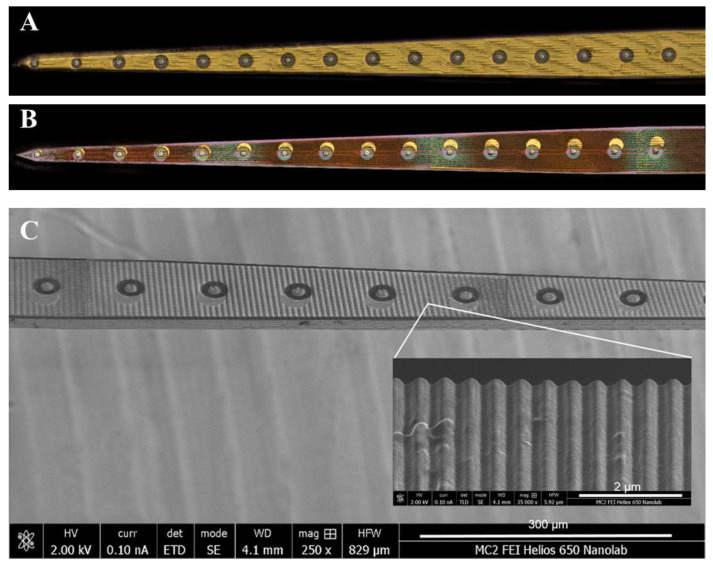
Keyence images of the top surface of NeuroNexus probes before (**A**) and after (**B**) focused ion beam etching. The nanopatterned grooves formed by focused ion beam etching diffract light, giving rise to a colorful appearance. (**C**) Scanning electron micrograph (SEM) image of the entire shank shown 250× magnification (scale bar is 300 µm). The inset depicts nanopatterned grooves at 35,000× magnification (scale bar is 2 µm).

**Figure 3 micromachines-15-00286-f003:**
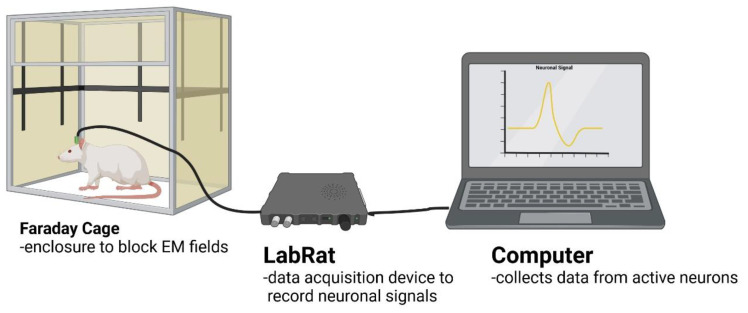
Diagram showing data pipeline from neural signals detected in the rat’s brain using the lab rat DSP and collection using SynapseLite via a laptop computer.

**Figure 4 micromachines-15-00286-f004:**
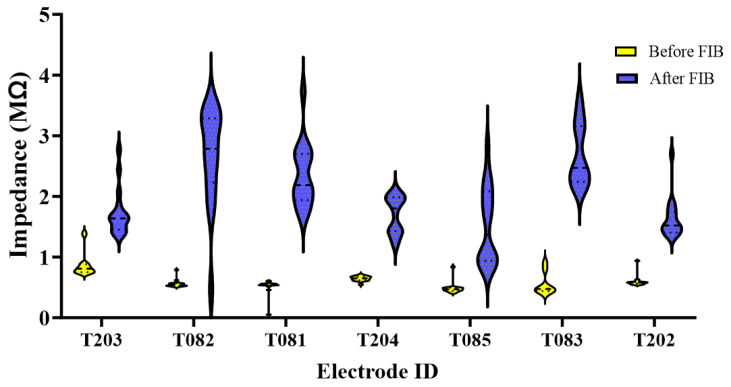
Violin plots of patterned probe impedances before and after FIB for probes that were used for neural recording.

**Figure 5 micromachines-15-00286-f005:**
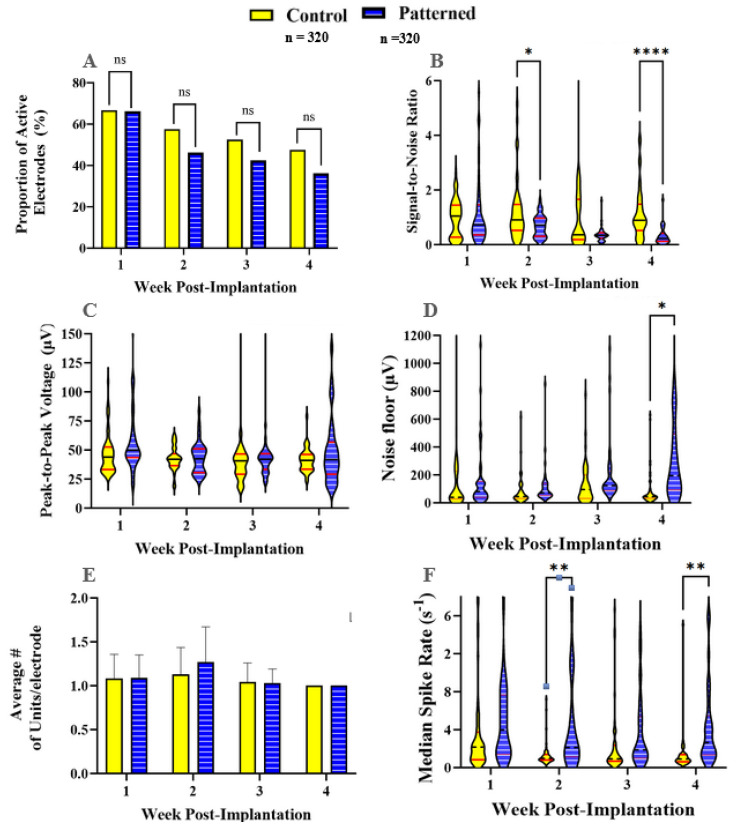
Recording metrics for control and nanopatterned electrodes across n = 320 total, with n = 80 weekly recording sites per experimental group for 4 weeks per experimental group (**A**) Bar graph showing the proportion of active electrodes. (**B**) Violin plot showing the signal-to-noise ratio. (**C**) Violin plot showing the peak-to-peak voltage. (**D**) Violin plot showing the RMS noise floor. (**E**) Bar graph showing the average # of units/electrode site. (**F**) Violin plot showing median spike rate. * = 0.01–0.05, ** = 0.001–0.01, *** = 0.0001 to 0.001 and **** =< 0.0001.

**Figure 6 micromachines-15-00286-f006:**
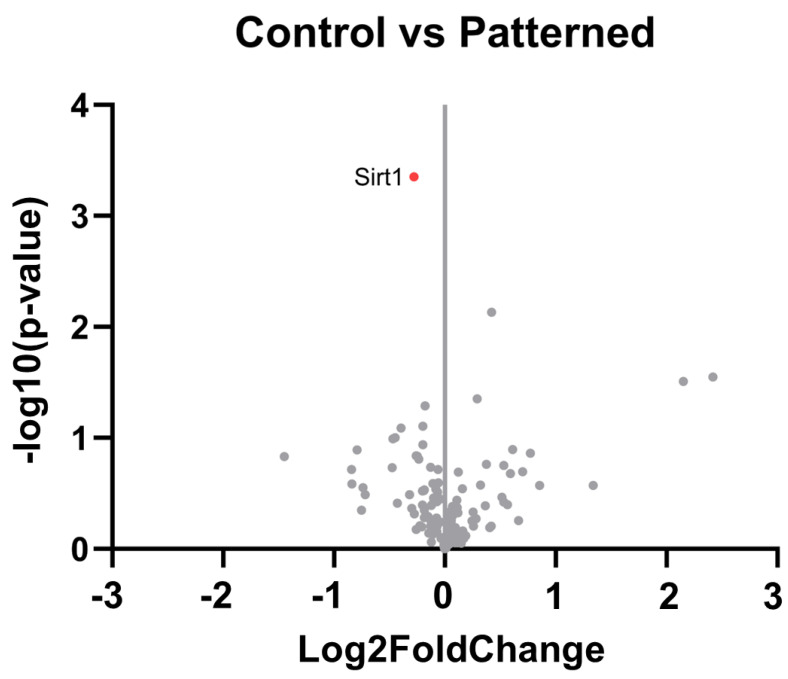
Differential expression between control electrode baseline vs. nanopatterned electrode at 4 weeks post-implantation. Genes that showed significant differential expression (padj < 0.1) are red and labeled. Genes that were not significantly differentially expressed are in gray. n = 6 animals per group.

**Table 1 micromachines-15-00286-t001:** Complete list of neuroinflammatory and oxidative stress genes of interest utilized in the study. Here, we list the 152 genes examined in rat brain tissue in this study using a combination of custom genes, preset genes from NanoString and housekeeping genes.

Custom Gene Panel			NanoString Preset Panel				Housekeeping Genes
*Aim2*	*Gsta1*	*Scd1*	*Abl1*	*Fas*	*Lpo*	*Sirt1*	*Hprt*
*Arc*	*Gsta2*	*Serpina3n*	*Ager*	*Fn1*	*Lrrk2*	*Sirt2*	*Rpl13a*
*Bdnf*	*Gstm2*	*Sod3*	*Aif1*	*Fos*	*Mapt*	*Slc8a1*	*Rps18*
*Blnk*	*Hmox1*	*Spp1*	*Akt1*	*Fxn*	*Mgmt*	*Snca*	*Sdha*
*C3*	*Il1b*	*Srxn1*	*Apoe*	*Gnao1*	*Mmp14*	*Sod1*	*Tbp*
*C3ar1*	*Il2rg*	*Tnfrsf1a*	*App*	*Gpr37*	*Mutyh*	*Sod2*	*Ubc*
*C4a*	*Irak4*	*Tnfrsf25*	*Atf4*	*Gsk3b*	*Ncf1*	*Src*	
*C5ar1*	*Irf7*	*Txnrd1*	*Atp13a2*	*Gsr*	*Nefh*	*Stx2*	
*Casp8*	*Itgam*	*Tyrobp*	*Atp7a*	*Gss*	*Ngfg*	*Tnf*	
*Ccl1*	*Keap1*	*Vegfa*	*Atrn*	*Gstp1*	*Ngfr*	*Tor1a*	
*Cd14*	*Lilrb4a*		*Bad*	*Gucy1b3*	*Nme5*	*Tpm1*	
*Cd36*	*Mmp12*		*Bcl2*	*H2-t23*	*Nol3*	*Trp53*	
*Cd45*	*Mpeg1*		*Bnip3*	*Hdac2*	*Nos1*	*Trpm2*	
*Cd68*	*Nfe2l2*		*Casp3*	*Hdac6*	*Nos3*	*Txnl1*	
*Cd74*	*Noxa1*		*Ccl5*	*Hgf*	*Nr4a2*	*Ubqln1*	
*Cd84*	*Nqo1*		*Ccs*	*Hif1a*	*Oxr1*	*Xbp1*	
*Clec7a*	*Nr2f6*		*Cdk2*	*Hspb1*	*Park7*		
*Ctss*	*Osgin1*		*Cln8*	*Htra2*	*Parp1*		
*Dock2*	*Osmr*		*Cybb*	*Idh1*	*Pdgfrb*		
*Ehd2*	*Prnp*		*Cycs*	*Il1r1*	*Pink1*		
*Ercc6*	*Psmb8*		*Ddit3*	*Il6*	*Pla2g4a*		
*Fcer1g*	*Ptgs2*		*Dnm2*	*Ins2*	*Ppargc1a*		
*Fcgr2b*	*Ptpn6*		*Ep300*	*Ipcef1*	*Psen1*		
*Gfap*	*Ptx3*		*Erlec1*	*Jun*	*Rela*		

## Data Availability

The raw data supporting the conclusions of this article will be made available by the authors on request.

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
