# Peer review of "In Vivo Characterization of Intracortical Probes with Focused Ion Beam-Etched Nanopatterned Topographies"

_micromachines, 2024, doi:10.3390/mi15020286_

Round 1

Reviewer 1 Report

Comments and Suggestions for Authors

The manuscript “In Vivo Characterization of Focus Ion Beam Etched Nanopatterned Microelectrodes to Improve Acute Recording Performance” by J. Duncan et al. presents results of the experimental comparison of “smooth” and patterned intracortical microelectrodes. Patterning of the microelectrodes was carried out using a focused ion beam (FIB) lithography. A distinctive feature of this work is a very detailed description of the equipment used and methods for preparing laboratory animals. In general, the obtained results can hardly be considered breakthrough, but they have some value for the further development of technologies for creating brain-machine interfaces. This work seems suitable for publication in “Micromachines”; however, some corrections and additional considerations are recommended to improve the quality of the manuscript.

1. The title of the article does not seem entirely appropriate. I don't see how “in vivo characterization” can “improve acute recording performance”. Also, the presented results do not show any remarkable influence of electrode pattering on the “recording performance” (Fig. 4), except occurrence of the Sirt1 gene (Fig. 5). Finally, еe structures formed on the electrode surfaces are quite difficult to attribute to the nanoscale (500 nm pitches).

2. The results of impedance measurements of smooth and patterned microelectrodes (Fig. 3) require a more detailed discussion with answers to the questions listed below.

a) Why, within the framework of the applied impedance measurement procedure, the patterning of the electrode surfaces leads to such a significant (2-3 times) increase in impedance? As follows from the description of the applied FIB technique, “etching was not performed within a distance of 3-10 μm from each recording site” (lines 117-118).

b) As follows from Fig. 1, the etched structures are fairly regular (non-stochastic). Why are patterned electrodes, in contrast to smooth ones, characterized by such a significant scatter of impedance values in the violin diagrams?

c) Why, with such a significant difference in the impedances of patterned and smooth electrodes, are their recording metrics (Fig. 4) close to each other?

3. The English writing needs correction of some inaccuracies and typos.

Comments on the Quality of English Language

Minor editing of English language is required.

Author Response

Please find Reviewer Comments in BOLD FONT and our response in italics.

1) The title of the article does not seem entirely appropriate. I don't see how “in vivo characterization” can “improve acute recording performance”. Also, the presented results do not show any remarkable influence of electrode pattering on the “recording performance” (Fig. 4), except occurrence of the Sirt1 gene (Fig. 5). Finally, еe structures formed on the electrode surfaces are quite difficult to attribute to the nanoscale (500 nm pitches).

We agree with the reviewer, and have changed the title accordingly to "In vivo Characterization of Intracortical Probes with Focused Ion Beam-Etched Nanopatterned Topographies". We have removed references to improving the performance. The reference to the nanopatterning was retained for the purpose of differentiating from the microscale features of the probes and microelectrode contacts.

2) The results of impedance measurements of smooth and patterned microelectrodes (Fig. 3) require a more detailed discussion with answers to the questions listed below.

a) Why, within the framework of the applied impedance measurement procedure, the patterning of the electrode surfaces leads to such a significant (2-3 times) increase in impedance? As follows from the description of the applied FIB technique, “etching was not performed within a distance of 3-10 μm from each recording site” (lines 117-118).

We have modified the text in the description of FIB etching to state that the file was designed such that all FIB etching would take place outside of that distance. However, as the FIB etching was taking place on microscale probes, some misalignment did take place, resulting in FIB etching on some contacts unintentionally. 

We have added the following text related to the impedance measurements to Section 3.1:

Violin plots of all respective values before and after FIB demonstrate an increase in impedance for some of the contacts following FIB treatment, which is likely attributed FIB-induced damage to microelectrode contacts. The large range of post-FIB impedance values measured correlates with the increasing misalignment in patterning along the length of the probe (Figure 2). Microelectrode contacts toward the tip are well-aligned, with all FIB exposure occurring outside of the contact. Toward the base, the slight rotational misalignment is magnified, resulting in FIB patterning on the microelectrode contacts. Since the thickness of the contacts is like the depth of nanogrooves, it may be the case that some of the microelectrodes were damaged, resulting in higher impedance values.

b) As follows from Fig. 1 (now Fig 2), the etched structures are fairly regular (non-stochastic). Why are patterned electrodes, in contrast to smooth ones, characterized by such a significant scatter of impedance values in the violin diagrams?

The significant scatter is believed to be related to the range of impact from FIB etching, ranging from no impact for well-aligned contacts to 100% damage for contacts with 100% FIB etching.

The text has been revised to reflect this possibility.  See section 3.1.

c) Why, with such a significant difference in the impedances of patterned and smooth electrodes, are their recording metrics (Fig. 4) close to each other?

Results from the impedance testing included every channel, without removing samples that the FIB process may have damaged.  However, recording studies removed animals that failed to record single units over the entire duration of the study (already mentioned in Section 2.3). 

No changes were made to the revised manuscript.

3) The English writing needs correction of some inaccuracies and typos.

Edits have been made throughout. We followed the recommendations of a grammar-checking program, Grammarly, to ensure all suggested edits were made correctly. Changes were minor throughout and were thus not noted in the revised manuscript.

Reviewer 2 Report

Comments and Suggestions for Authors

The manuscript on "In Vivo Characterization of Focus Ion Beam Etched Nanopatterned Microelectrodes to Improve Acute Recording Performance" is an interesting study at least for a specialized audience although not yet a specific conclusion can be drawn on the exact benefits of the design modifications. Overall, the study is clearly described besides the design and processing details of section "2.1 Neural Probe Manufacturing", which for clarity should be improved by a process drawing of the steps or at least a schematic on the actual relevant design features in an technical drawing rather in text. Line 91-129 are hard to follow. Please consider adding a drawing (maybe as part of the paper in a supplementary) of the pattern overlay between the electrode arrangement of the probes and the nanopattern arrangement.  Also an overview image of how the probe is mounted for the FIB process could be helpful.

Author Response

Please find Reviewer Comments in BOLD FONT and our response in italics.

The manuscript on "In Vivo Characterization of Focus Ion Beam Etched Nanopatterned Microelectrodes to Improve Acute Recording Performance" is an interesting study at least for a specialized audience although not yet a specific conclusion can be drawn on the exact benefits of the design modifications. 

  • We thank the reviewer for the positive assessment of the work and the keen understanding of the progress made thus far in a much longer study to come.

Overall, the study is clearly described besides the design and processing details of section "2.1 Neural Probe Manufacturing", which for clarity should be improved by a process drawing of the steps or at least a schematic on the actual relevant design features in an technical drawing rather in text. Line 91-129 are hard to follow. Please consider adding a drawing (maybe as part of the paper in a supplementary) of the pattern overlay between the electrode arrangement of the probes and the nanopattern arrangement.  Also an overview image of how the probe is mounted for the FIB process could be helpful.

  • We have added a new Figure (now Figure 1) to the Methods Section in Section 2.1 to address the reviewer's points.  The new figure shows the process steps and a photo of the probe mounted to the aluminum stub for FIB.  All subsequent figures have been renumbered.
  • Additional minor edits to the text in Section 2.1 were made to better describe the process and call attention to the new Figure 1.  Changes made with blue font.

Reviewer 3 Report

Comments and Suggestions for Authors

This manuscript studied the effect of neural probes with nanopatterned grooves etched by focus ion beam etched on improving acute recording performance. The experimental method is novel, and the manuscript is written in a standardized and detailed manner, providing a good reference for the study of novel nanoscale topographies on neural probes.

Author Response

We thank the reviewer for the overall positive assessment of our manuscript.  Without any specific items to address, we made no changes to the manuscript in response to Reviewer #3.